# Educational Needs in Oncology Nursing: A Scoping Review

**DOI:** 10.3390/healthcare10122494

**Published:** 2022-12-09

**Authors:** Silvia Solera-Gómez, Amparo Benedito-Monleón, Lucía Inmaculada LLinares-Insa, David Sancho-Cantus, Esther Navarro-Illana

**Affiliations:** 1Hospital Francisco de Borja, 46701 Gandia, Valencia, Spain; 2Faculty of Psychology, University of Valencia, 46000 Valencia, Valencia, Spain; 3Faculty of Medicine and Health Sciences, Department of Nursing, Catholic University of Valencia, 46600 Valencia, Valencia, Spain

**Keywords:** cancer nursing, cancer care, continuous learning, education gaps, educational intervention

## Abstract

Care in oncology requires both technical and psychosocial skills by nursing staff, so continuous learning is necessary. Evidence suggests there are some educational gaps in oncology nursing staff, and continuing educational interventions have been effective in overcoming these deficiencies. Aim: to determine the basic educational lines that a continuous training program should have for oncology nurses. A bibliographic review study was carried out in two phases from October 2020 to January 2021. In a first phase, the main databases were analyzed: PubMed, Web of Science, Dialnet and Medline, following the PRISMA methodology; and subsequently, an analysis of the most important thematic nuclei that a training program in cancer nursing should contain. The DAFO matrix and the Hanlon prioritization method were used. Four competencies that every oncology nurse should have were described: communication, coping, self-direction of learning and technical health. The thematic contents that a training program should contain were then determined, and aspects such as stress prevention and burnout, adequate communication with patient and family, and continuous educational and technical skills were considered. The results found suggest that there are deficiencies in the education of nursing staff. Continuing education programs are effective in supplementing them. They should develop the four skills described in the results section.

## 1. Introduction

Cancer treatment places a great physical, psychological and social burden on the patient [1,2], which may lead to greater psychological stress, often manifested by post-traumatic stress syndrome, depression or anxiety disorders [2,3]. These pathologies have been directly and inversely related to the patients’ quality of life, who make greater use of healthcare resources, show greater difficulties in decision making, have lower rates of treatment adherence, and are less satisfied with the care received [4,5], showing a higher need for care, attention and dependence on oncology nurses.

Due to the rising incidence of cancer, the demand for specialized and emotionally trained oncology nurses continues to increase [6,7] since they have the advanced education in care and leadership skills, and in the management of healthcare resources for the comprehensive care of patients [8], for whom the mere presence of nursing staff generates positive effects on recovery and self-care [9].

For oncology nurses, adequate education is one of the most effective measures to reduce stress levels and improve self-efficacy and work performance [10,11,12,13]. Evidence points to the effectiveness of intervention programs [14] based on the development of certain competencies. In order to determine these competencies, the opinion of the nursing staff will be essential since they know what their educational needs are [3,15,16,17,18,19]. However, the literature also highlights a wide variety of competencies that can be improved. The National Institute for Health and Care Excellence (NICE) guidelines for cancer management establishes that assessing the physical, psychological, spiritual, social and even economic needs of patients is highly important [16,20,21,22], and these guidelines contemplate the figure of the health professional as a care provider and guarantor of patients’ rights [23,24,25]. Nurses, together with oncology doctors, accelerate the provision of health care, improve ongoing care, minimize the symptoms and adverse effects of treatment, help patients to face their fears, improve their relationship with their environment or detect situations of special fragility [26]. Along these lines, the National Cancer Institute (USA) included training to improve communication skills in relation to the palliative oncology patient in the nursing curriculum [27,28,29]. Among the psychosocial skills, communication is the backbone of the relationship between health professionals and patients and their families [30], making adequate communication essential from the early stages in regard to the diagnosis, prognosis, mental distress and treatment options [25,31,32]. Adequate communication skills in oncology nurses are also linked to greater adherence to treatment and better psychosocial functioning of the patient [18,33]. In addition, nurses often “translate” the information provided to patients into understandable language and help the patient and family to learn to manage the disease process [34].

Aspects such as active listening and empathy are also basic tools for oncology nurses [35,36,37,38] and help them to deal with complex situations [18]; hence, the Nursing Interventions Classification (NIC) already considers numerous activities related to communication, such as active listening, promotion of socialization or support for the family.

Moreover, in order to provide adequate care and be capable of dealing with complicated situations, oncology teams need to be able to carry out interventions based on coping with emotional stress and workload [4].

The literature also highlights the importance of self-learning competencies in nursing education as nurses need to constantly improve their work process given the specific characteristics of the oncology patient and their family [39]. There is growing interest in how the patient feels, how the disease process influences the provision of care, or the barriers that prevent professionals from providing higher-quality care [29].

It is also essential to raise awareness among nurses about the need for continuous education based on the development of the necessary competencies to provide quality care [40]. In the past, oncology nurses used to receive education or orientation in specific topics related to their tasks in the workplace [18,41]. Nowadays, this education has largely ceased to be provided and many nurses lack the specific knowledge required to work in oncology units [23] and there is a need to deepen those aspects that improve competencies in the management of complex clinical situations [6]; hence, numerous intervention programs designed to promote greater autonomy and patient management have been implemented in oncology.

Psychosocial skills, coping strategies, self-directed learning and professional clinical training help nurses to face professional challenges adequately [6,42]. Therefore, the aim of this literature review was to identify the basic lines of education that a continuing education program for oncology nurses should follow, according to the evidence available.

## 2. Materials and Methods

### 2.1. Thematic Core of an Educational Program: Process Design

A literature review was conducted on the skills of nursing staff working with cancer patients and the educational needs that emerged from the shortcomings detected. The Sidani and Braden steps [43] for educational programs design were followed.

The second phase consisted of designing the thematic core for an educational proposal for oncology nurses by determining the specific and non-specific elements to be considered.

### 2.2. First Phase of the Study: Literature Review

The objective of the search was to determine the educational needs of oncology nurses following the Preferred Reporting Items for Systematic Reviews and Meta-Analyses (PRISMA) guidance [44]: (1) identification of the problem, (2) literature search, (3) evaluation and analysis of the main results.

The scoping search was conducted in the PubMed, Web of Science, Dialnet and Medline databases, between October 2020 and January 2021. A combination of the following keywords was used: needs, education, training, nursing staff, oncology nurse, combined with the corresponding Boolean operators. Table 1 shows the search strategy and the results obtained.

As shown in Figure 1, systematic reviews, meta-analyses, randomized clinical trials, quasi-experimental and qualitative studies were eligible. Studies published in the last 10 years, in English/Spanish, available in full text, which addressed the educational needs of oncology nurses were included. Patients seen in special units such as emergency departments, operating rooms or mother and child units; population under 18 years of age; and cancer patients not looked after by nursing staff; as well as studies related to the main carers of cancer patients, were excluded.

839 articles were identified by the search strategy. The Mendeley software was used to organize the search results. After eliminating duplicates, 284 articles were selected. A total of 141 articles were selected for full-text reading and 80 were finally excluded as they did not meet the requirements of the study. Two reviewers independently screened publications for eligibility and all authors approved the final sample of articles, which consisted of 61 studies.

A content analysis of the articles was performed through critical reading [43]. The articles were coded and grouped into categories, including education or stress for example, and according to similar topics [44].

### 2.3. Second Phase of the Study: Thematic Core of the Education Program

For the analysis of the education needs for oncology nurses, a SWOT analysis was performed. The strengths and weaknesses detected in the literature review were analyzed (Table 2) to complement the identification of needs from the search. The SWOT analysis is a business analytical technique that analyzes the main elements that make up a product or service (strengths, weaknesses, opportunities, and threats) in order to establish strategies for improvement [45].

The Hanlon method was used to rank the different topics obtained in the search in order to build the thematic axis of the program (Table 3). The method establishes 4 categories to prioritize more objectively when making decisions related to resource management: size of the problem (S), seriousness or magnitude (M), effectiveness of potential interventions (E), and feasibility of the program (F) [46,47]. The importance of each component is reflected in its corresponding scale. Based on the results obtained, two educational axes for oncology nurses were proposed.

## 3. Results

### 3.1. Phase I: Literature Search

839 articles were obtained and 61 of those were selected for further analysis. A total of 29 were quasi-experimental pre-post-test studies in which the aim was to evaluate the efficacy of an education or intervention program There were 20 qualitative studies, 5 systematic reviews, 5 randomized clinical trials and 2 cross-sectional descriptive studies. A total of 35 of the articles had been published in the last 5 years, 9 in 2017, 5 in 2018, 7 in 2019, 3 in 2020 and 1 in 2021.

The search showed evident existing education gaps in oncology nursing. For the content analysis, the 61 articles were grouped into the following categories: (a) communication, (b) coping, (c) self-directed learning, (d) technical competencies. The most relevant results of each category are gathered and discussed on Table 4.

#### 3.1.1. Communication Competencies

Communication turned out to be essential in professional–patient interaction to the extent that it influenced the way of coping with the disease. Davis analyzed the complexity of dealing with the oncology patient from a phenomenological perspective and considered the qualities an oncology nurse should possess [23]. Barerjee implemented a program aimed at improving communication skills among nurses working in oncology units. Aspects such as stress management or decision making also showed to be effective in the psychosocial performance of the professionals [31]. Barth and Lannen analyzed the variables in the communication process with the patient through a meta-analysis through the analysis of 13 clinical trials, 10 of them randomized, and concluded that measures to enable nurses to improve their communication skills needed to be implemented [4].

A series of studies that explored aspects such as empathy or active listening, as well as the influence these had on the interaction with the patient and their family, were found. Through a qualitative study, Flocke and colleagues proposed detecting barriers in regard to counselling and the manner of treating the patient (lack of knowledge), and the main topic that emerged from their study related to nurse decision making [22]. Rohani explored the importance of empathy from the nurses’ perspective, and concluded that clinical empathy should be considered one of the essential competencies in this group [47]. Cara and colleagues sought to determine the needs for active listening in oncology nurses and identified communication difficulties reported by patients and by the nurses themselves [46]. Leal-Costa, on the other hand, sought to provide new knowledge in relation to communication skills in nursing and perceived self-efficacy, and concluded that communication skills can positively affect self-efficacy [48].

The following is a series of studies in which intervention programs related to communication skills were considered. Watson implemented a psychoeducational program by nurses caring for prostate cancer patients to ascertain the existing shortcomings related to psychosocial interaction [38], while Eid [42] or Mojrad [49] analyzed through a quasi-experimental study the effectiveness of a program aimed at improving the ability of nursing staff to communicate bad news, finding significant results in the medium and long term. Badger analyzed a sample of cancer survivors to identify the most effective interventions in order to improve psychological quality of life [50]. It was a psychosocial information program aimed at patients, although the findings were not consistent due to the small sample size. On the other hand, Wittenberg assessed the impact of an online educational program based on communication in nursing students in relation to their attitudes, knowledge and expectations towards the oncology patient as a preliminary step to the design of intervention plans for oncology nurses [18,51]. Finally, Borneman demonstrated an improvement in the nurses’ communication when dealing with oncology patients after implementing an educational program [19], as did Walczak [1].

After all the aforementioned, the importance of the correct use of communication as a tool for interaction with the patient and their family seems clear. It has been shown to be effective in reducing levels of anxiety or perception of insecurity in patients, which is why gaining these skills is essential.

#### 3.1.2. Coping Competencies

Coping strategies and their effectiveness, or the prevention of work-related stress, are some of the topics that emerge from this analysis. De Carvalho identified the feelings of nurses working with terminally ill cancer patients as a starting point to better understand the existing psychosocial deficiencies in coping with the relationship with the patient [2]. One of the main difficulties lies in accompanying the patient in the process of dying. Through a literature review, Pongthavornkamol approached the influence of nurses on oncology patients in terms of coping and concluded that there was not a specific protocol for detecting the needs of the patients [52].

With respect to the behavior of the professionals in relation to the patient and family, Harden suggested improving knowledge, attitudes and said behavior through an educational program and showed its effectiveness [53]. Fuoto looked into the attitudes of nursing staff towards the care of the family of the cancer patient and identified, through focus groups, the main demands of the nurses, among which were trust or security and a communicative relationship [13]. Petersen analyzed the competencies that pediatric oncology nurses should possess and concluded that assertive communication and the management of stressful situations were two fundamental aspects [28]. On the other hand, Schwappach suggested, through a qualitative narrative study, the use of a useful approach for helping oncology nurses to deal with patients and their families [39].

Concerning the perception of work stress and the prevention of burnout, Wenzel established coping and loss in relation to oncology patients and their families as the main categories to be considered in an educational program [14]. Park et al., through a descriptive study, identified signs of burnout in oncology nurses, although the percentage of cases was not higher than in other groups of nursing professionals [54]. In a study by Poulsen, categories such as lack of social support or emotional demands emerged when analyzing the reasons underlying occupational stress through semi-structured interviews [29].

Another underlying variable was the professional’s experience of the clinical process in oncology. Thomas analyzed the possible situations that could cause suffering or pleasure to oncology nurses and concluded that the main reason for stress and dissatisfaction was related to coping with patient death [55]. Vila proposed defining a new oncology nursing role to guide breast cancer patients and, through qualitative methodology, demonstrated that this role seemed useful for helping patients in problem solving, developing group work skills, and coping with burnout [26].

With reference to the intervention programs reviewed in the different studies, findings related mainly to the adequate management of work stress stand out. In a recent study, Bozorgnejad tried to determine the effects of a social network educational program for nurses in relation to stress management through a non-randomized clinical trial. Stress reduction as well as an improvement in self-efficacy were observed [10]. Likewise, Poulsen and colleagues proposed a program to try to reduce the stress suffered by oncology nurses in which they also attempted to evaluate the dimensions of professional stress [29].

After analyzing the texts, it was seen that there are educational deficiencies, in particular, when dealing with conflict resolution and when facing complex and stressful situations when interacting with patients and families.

#### 3.1.3. Self-Directed Learning Competencies

Brixey reflected on the design and implementation of self-assessment tools for oncology nurses to identify their own educational needs as one of the core issues prior to any type of educational intervention [16]. Dos Santos, in turn, reflected on the importance of continuous education in oncology nursing through semi-structured interviews [36]. Santana-Padilla investigated the educational deficiencies of personnel working in an ICU through a phenomenological study and concluded that there were shortcomings in undergraduate education prior to starting work, as well as in continuing education [25]. Continuing with the educational needs analysis, Stanciu used a needs assessment tool to evaluate oncology nursing interventions in relation to prostate cancer patients, and their program was shown to be effective in the self-management of the oncology patient [32].

With reference to the implementation of intervention programs, a randomized clinical trial by Henoch stands out in particular. It aimed to determine the effects of an educational intervention focused on the perception of nursing confidence in relation to communication with oncology patients, and the short-term results showed an improvement in the confidence of the nurses and the importance of education for middle management [56].

After analyzing these studies, the underlying educational needs for oncology nurses are related to the detection of needs by the professionals themselves. Therefore, the main conclusion is that nurses should undergo continuous education in order to adequately carry out their work and this learning should be self-directed, i.e., learning autonomously and according to their needs.

#### 3.1.4. Technical Competencies

This section includes studies about technical skills such as pain or general symptom management in the oncology patient. A systematic review by Borneman described the barriers for the correct management of cancer pain among nurses and concluded that there were not many studies that assessed that type of issue [19], and Vila studied the adjustment to the disease in a group of women with breast cancer through an intervention directed by nurses [26]. Along the same lines, Korber demonstrated the efficacy of a program to strengthen the self-care competence at home in a group of women with cancer, guided by oncology nurses [34].

Some studies suggest the benefit of the implementation of intervention programs for nurses in this regard. Zaider proposed an intervention to help oncology nurses to manage complex situations [37]. Meanwhile, Gustafsson investigated whether a theory-based educational intervention could modify knowledge and attitudes towards pain management in oncology patients and saw a reduction in both pain and fatigue, as well as an improvement in the nurses’ knowledge [24].

After analyzing these texts, the educational needs for oncology nurses that can be highlighted in terms of technical competencies are: pain management in the clinical context and self-care management for patients and family.

Implications for practice.

### 3.2. Phase II: Educational Proposal

#### 3.2.1. Definition of the Educational Focus

A set of educational needs emerged from the literature search (Phase I) and an educational proposal was developed (Phase II) based on them.

Through the SWOT analysis, the main thematic core to be included in the educational proposal was communication training. Subsequently, and as the last step in defining the objectives and contents of the program, the educational needs were prioritized. The Hanlon method was used for this purpose (Table 4). The scores obtained in each of the four dimensions of this method were as follows: in terms of the size of the problem, and on a scale of 0–10, a score of 8 was given to the category “communication skills”, according to the importance reflected in the literature. The category “technical skills” received a score of 5/10. The second dimension was the seriousness or magnitude, also on a scale of 0–10. The resulting score for the “communication skills” category was 8/10, compared to “technical skills”, which was 7/10. The third dimension was the potential effectiveness of the intervention, on a scale of 0.5–1.5, with the two categories above receiving the same rating: 1, while feasibility constituted the fourth dimension and both categories also scored 1 point on a scale of 0–1.

#### 3.2.2. Educational Proposal for Oncology Nurses

From the results obtained in the priority analysis, two major educational needs were identified according to the scores given. The first educational need would be the improvement of cross-sectional meta-competencies related to communication skills, coping abilities and self-directed learning, whereas the second would be the improvement in technical competencies related to specific aspects of the approach to clinical problems.

Communication with the patient and family has been shown to be an essential component that influences the quality of life of the family unit [66], so these skills will certainly be crucial, as highlighted by some authors such as Zaider [37]. Assertiveness, empathy or non-verbal communication become basic skills in the interaction with the patient and family and have been shown to be effective in reducing patient anxiety [32]. In Zaider’s study, nearly 40% of the nurses who participated reported the retention of specific skills six months after the end of the intervention [37]. Hendricks-Jackson observed a reduction in patients’ psychological stress levels after a psychoeducation-based intervention [57] and, after developing and evaluating a communication skills training program for nurses, Rohani was able to show a direct relationship between empathy and self-efficacy in nurses [47].

The role of nurses in the use of communication skills was considered “essential” to achieve positive outcomes [67]. There is evidence that communication in general, and particularly in more complex situations, is an area that needs to be developed to minimize errors or improve health outcomes [25]. Nurses recognized existing communication barriers with oncology patients and their families [37,68] and confessed their own shortcomings in relation to issues such as effective active listening or empathy in these situations [41,48,51]. Some studies have confirmed that empathy is a natural element in the context of oncology patient care, and have also emphasized the need to improve communication skills through education and continuing nursing education [2,42] and that it is possible to improve health outcomes following educational interventions based on psychosocial communication skills, including empathy [35,69,70].

In the specific case of end-of-life care or palliative care, communication becomes fundamental. Anxiety in the face of death may trigger defense mechanisms in the individual such as denial or avoidance as a means of trying to distance oneself or manage fear [31]. In addition, the lack of information was shown to be a factor that produces anxiety due to the insecurity and uncertainty it generates in the patient and family. Along these lines, interventions aimed at improving communication with advanced-stage cancer patients have focused on the impact of communication separately, i.e., on the patient or the professional, without considering the surrounding environment. In fact, few studies have analyzed the role of caregivers or the family in this communicative process [1]. There is evidence that intervention programs aimed at improving the communication skills of oncology professionals significantly helps with coping with managing bad news and with approaching palliative care [56,58].

With regard to coping abilities, according to Trevisani, almost 50% of the participants in educational programs reported not having received any coping-skills training and 30% did so sporadically [20]. In a study with a sample of nurses working in a day hospital, three types of factors were found to be responsible for deficiencies in professional coping: factors related to the working conditions, interpersonal factors, and communication problems within the team [9]. Other factors related to the previous ones which were also highlighted were those referring to the emotional demands of patients or relatives. The importance of adequate communication and management of the professional’s own emotions in these circumstances was also pointed out. A workshop on the management of work-related stress was carried out in that study and its evaluation was generally positive.

Some authors reported significant improvements in coping with the complex situations that occur in relation to cancer patients and their families, showing a reduction in the main stress indicators in professionals after the intervention [6,58]. Aspects related to the management were found to be important as managers complement the professionals’ coping systems. In other words, stress prevention policies implemented by the respective management teams in the centers influence the care delivery system [54].

Self-efficacy expectations have been directly related to care delivery as they are considered relevant in predicting the individuals’ ability to cope with stress when delivering care [12]. Improvements in self-efficacy following educational intervention have been demonstrated.

In relation to self-learning, and unlike other fields such as intensive care (ICU), there is no consensus in Spain, for example regarding the curricular profile, roles or activities of an oncology nurse. With regard to ICU, some tools have been proposed to homogenize practices throughout Europe based on the proposals of the Critical Care Nursing Association [25]. Nonetheless, there is a clear lack of training in communication skills in basic and specialized nursing education [42] and there are neither unified nursing protocols for the management of oncology patients nor advanced practice nursing roles developed in oncology [9]. The Spanish Oncology Nursing Society (SEEO) has highlighted the need to establish a competency-based framework and the main recommendations are summarized in three basic pillars: patient assessment, clinical management and counselling [26]. Thus, the roles of oncology nurses should include assessment of the patient’s individual needs, education, communication, coordination between the different levels of care, and implementation of effective transitions in the course of the disease, as well as evaluation of the consequences of this whole process for patients and families [20]. In line with this, the SHARE project (Sessions for Interdisciplinary Nursing Analysis and Review) proposed by the SEEO was developed in response to the need to share the advanced practices in oncology nursing that were already being carried out in some centers. The figure of the nurse case manager was implemented as a guarantor of continuity of care by facilitating the patient’s ability for self-care and increasing patient confidence [26].

Different authors identified the lack of specific education in fields such as oncology nursing or intensive care, and proposed the implementation of regulated educational programs. This need for continuing education has emerged not only from scientific societies but also from the demand of the professionals themselves [25]. Educational programs that develop competencies contribute to the provision of high-quality care, as well as to the systematization of nursing interventions [59].

There are limitations encountered by oncology nurses when attempting to update their knowledge and the literature highlights the high workload and the specificity of this type of patient, which requires constant knowledge and skills renewal [34,60]. These shortcomings were also observed when implementing a specific palliative care program [61] and the effects seen in the professionals were lack of self-confidence and adequate education in palliative care, as well as high levels of stress and burnout syndrome [14].

In the COMFORT program developed by Fuoto and Turner, the effects of communication with the terminally ill cancer patient were analyzed and significant improvements were seen in terms of professionals’ satisfaction and confidence in providing care [13]. The professionals also reported feeling more prepared and with a greater awareness of everything surrounding the patient and the process of disease. This was made possible by allowing nurses to practice communication in a controlled environment.

Park developed and implemented the BRAVO-NTP program to provide adequate education and resources to healthcare professionals based on the psychosocial and physical demands in oncology [54].

Regarding the technical competencies, in the case of pain management, there is evidence that supports the effectiveness of intervention programs led by oncology nurses [19]. The implementation of this type of intervention led to an increase in patients’ knowledge and a change in their attitude towards the disease. Secondarily, these interventions reduced the levels of pain perceived by the patient and the results obtained seemed to improve in direct proportion to the duration of the program. Furthermore, these results correlated with quality of life and patient satisfaction with the care received [19,24].

It was also highlighted that in order to improve the effectiveness of these interventions, the level of knowledge of the professionals needed to be continuously improved [62]. Less than 14% of the professionals were aware of their patients’ preferences concerning pain management or the place where they wanted to die [1].

## 4. Discussion

The aim of this study was to design an educational program to improve communication skills in oncology nurses. The first step consisted of a search for existing evidence, and communication, coping skills and self-directed learning were identified as the core concepts on one hand, and technical competencies on the other. In the literature, the first thematic axis appeared to be more relevant than the second due to the prevalence of relational aspects over purely clinical or technical ones.

In a quasi-experimental study on the commitment of nursing staff, improvement in patient satisfaction and emotional state was observed after an intervention carried out by nurses, compared to another intervention developed by medical staff [54,63]. Another review [64] revealed an improvement in the quality of life reported by patients, and better cost-effectiveness rates when fewer tests and supplementary tests were performed, in programs implemented by nursing professionals.

Psychosocial resources act as predictors for differential response to the intervention [54,65,68]. Parameters such as anxiety or depression also improved after nursing intervention. In addition, a reduction in chemotherapy symptoms and a positive impact of effective communication were seen, enabling a better understanding of the indications given to the patient [58].

Banerjee also found a significant improvement in nursing knowledge after implementing a program [31]. Thomas identified that patients felt the need to ask themselves deeper questions about their existence after being diagnosed with cancer, and that the nursing staff lacked sufficient education to address those questions [55]. Nursing professionals recognized aspects such as warmth, cordiality and the ability to empathize as elements favoring communication. Likewise, as main difficulties in the communication process, those related to the lack of education in that front were highlighted [38].

In the different programs reviewed, significant differences in knowledge, attitudes and behaviors of the professionals were outlined, in addition to obtaining good evaluations after the end of the interventions [5,23,30,61]. Intervention programs based on religious or spiritual components revealed effective and lasting results regarding the management of the meaning of life in cancer patients [28].

With regard to coping, the programs reviewed showed their efficacy in reducing stress [19,29] and burnout [11,57] suffered by oncology nurses. A direct relationship was also observed with self-learning [20] as these programs gave the professional greater autonomy in the search for resources.

As per the technical competencies [19,22], an increase in the professionals’ self-confidence in relation to the improvement in the non-technical competencies mentioned above was seen. Thus, improvements in relational skills contributed to a greater perception of security in nurses.

The main limitation of the present study was its excessively theoretical approach. In this sense, and as for the practical implications, longitudinal research is proposed in which monitoring or follow-up can be carried out to complete the information collected, as well as the design and implementation of a specific educational program based on the thematic core described in this review. Nonetheless, the novelty of this educational program lies in the approach to the specific skills to be worked on, since most of the interventions seen in the literature are much more general.

## 5. Conclusions

The cancer burden continues to grow globally as does the demand for healthcare resources; it is, therefore, essential to provide nursing professionals with the appropriate tools to deliver quality care.

A great deal of research has demonstrated the effectiveness of formative programs for the professional education training of nurses and multiple factors have been involved in the response to the different intervention programs revised.

The effectiveness of nursing interventions was found to be significant when educational and psychosocial approaches were used. Psychosocial interventions were shown to be effective in improving the psychological distress of cancer patients and their quality of life, specifically in those brief intervention programs that incorporated psychoeducation or counselling where communication with the patient was the central element.

There is evidence supporting the effectiveness of intervention programs in cross-sectional competencies, so the goal of this study was to propose lines of education for oncology nurses according to their needs. For this needs analysis, a literature review was carried out and the thematic cores proposed for the continuous education of oncology nurses are communication, coping, self-learning and technical competencies. This study constitutes a tool to improve communication skills in complex contexts such as those related to the field of oncology.

## Figures and Tables

**Figure 1 healthcare-10-02494-f001:**
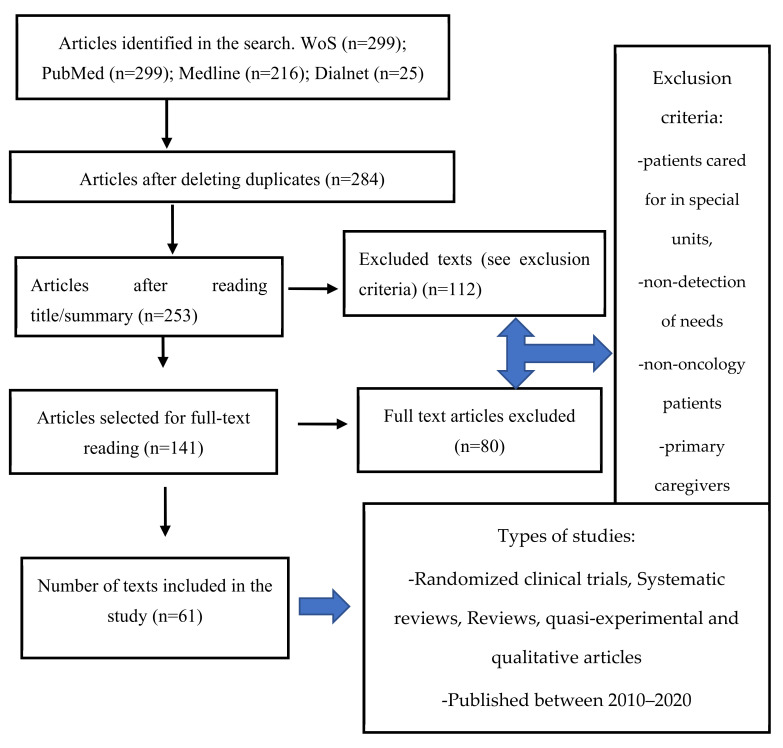
PRISMA flow diagram for study selection.

**Table 1 healthcare-10-02494-t001:** Search strategies in the different databases.

Search Strategies	Pubmed	Medline	Dialnet	WoS *
((“nursing”[Subheading] OR “nursing”[All Fields] OR “nursing”[MeSH Terms] OR “nursing”[All Fields] AND (“neoplasms”[MeSH Terms] OR “neoplasms”[All Fields] OR “oncology”[All Fields])) AND (“education”[Subheading] OR “education”[All Fields] OR “teaching”[All Fields] OR “teaching”[MeSH Terms])	172	136		166
(“nursing”[Subheading] OR “nursing”[All Fields] OR “nursing”[MeSH Terms] OR “nursing”[All Fields] AND ((“neoplasms”[MeSH Terms] OR “neoplasms”[All Fields] OR “oncology”[All Fields]) AND (“patients”[MeSH Terms] OR “patients”[All Fields] OR “patient”[All Fields]))) AND (“education”[Subheading] OR “education”[All Fields] OR “teaching”[All Fields] OR “teaching”[MeSH Terms]) AND (“2010/11/28”[PDat]: “2020/11/24”[PDat])	127	80		133
Enfermería AND necesidad AND formación			25	

* WoS = Web of science.

**Table 2 healthcare-10-02494-t002:** SWOT analysis of education needs for oncology nurses.

SWOT Analysis	Strengths	Weaknesses
Internal analysis	- Previous education - Experience	- Little education in emotional management and conflict resolution. - Resistance to change. - Staff motivation problems.
	Opportunities	Threats
External analysis	- Updating of education - Increased personal motivation - Increased satisfaction	- Difficulty in making time and resources available for education.

**Table 3 healthcare-10-02494-t003:** Prioritization of educational actions according to the needs detected. Hanlon Method.

PROBLEMS DETECTED [(S + M) × E × F]	1	2	3	4
Communication, Coping and Self-learning Competencies	8	8	1	1
Technical Competencies	5	7	1	1

**Table 4 healthcare-10-02494-t004:** Overview of included evidence.

Author/Year	Study Design	Purpose	Primary Outcomes
Walczack et al. 2014 [1]	Randomized clinical trial	Evaluation of a program to improve communication skills in oncology nurses	Effectiveness of the program to improve communication skills
Alencar et al. 2017 [2]	Descriptive	Identify the feelings of oncology nurses in palliative care	Urgent need to create support groups among nursing professionals
Dastan et al. 2012 [3]	Randomized clinical trial	Assess the effects of a psychological intervention on levels of adjustment to breast cancer	No significant differences were shown
Barth et al. 201 [4]	Meta-analysis	Assess communication skills training in oncology	Working on communication skills is important to improve oncology outcomes
Head et al. 2016 [5]	Quantitative–qualitative study	Design and implement an interdisciplinary curriculum teaching team-based palliative care in oncology	Success of one university’s effort to design and implement an interdisciplinary curriculum teaching team-based palliative care in oncology
Britt et al. 2012 [7]	Quasi-experimental	Evaluate the personal and organizational impact of an educational intervention on the stress of healthcare team members	Stress and its symptoms are problematic issues for hospital and ambulatory clinic staff as evidenced by baseline measures of distress
Burton et al. 2010 [8]	Descriptive	reunir información preliminar sobre la viabilidad de aplicar un programa de formación en resiliencia psicosocial en grupo (REsilience and Activity for every DaY, READY) en un entorno laboral, y evaluar si el programa podría promover el bienestar	Los participantes valoraron muy positivamente el programa y los materiales. Estos resultados indican que el programa READY es factible de aplicar como programa de formación en grupo en un entorno laboral para promover el bienestar psicosocial.
Reñones et al. 2016 [9]	Descriptive	Program for the improvement of the oncology patient (SHARE project)	Fundamental aspects of oncology patient care are highlighted
Bozorgnejad et al. 2021 [10]	Non-randomized clinical trial	Determine the effects of a social network training program for nurses in relation to stress management	Reduced stress and increased nursing self-efficacy
Aranda et al. 2012 [11]	Randomized clinical trial	Evaluate the effects of an intervention program to reduce different symptomatology in oncology patients based on their needs	Chemo-ed program proves effective in the management and control of oncology patient distress
Penagos et al. 2020[12]	Quasi experimental	Determine the effectiveness of a program to prevent overload in the primary caregiver of the oncology patient	Effectiveness of the motivational approach in educational programs in oncology nursing
Fuoto et al. 2019 [13]	Descriptive	Improve the safety and satisfaction of oncology nurses in relation to communication skills in palliative care	No significant differences were observed
Wenzel et al. 2011 [14]	Descriptive	Determine the facilitators and barriers to oncology patient management and extract the main components that an intervention program should have	Emerging categories: coping and loss
Houck et al. 2014 [15]	Descriptive	Present an educational intervention to help nurses cope with grief and compassion fatigue	----
Brixey et al. 2010 [16]	Descriptive	Implement and describe a self-assessment tool for oncology nurses	Utility of the self-assessment tool to detect needs in oncology nurses
Baer et al. 2013 [17]	Descriptive	Describe the impact of communication techniques on oncology patients	Communication appears as one of the basic skills in the interaction with the patient
Wittenberg et al. 2018 [18]	Descriptive	Assess the impact of an online communication-based training program in nursing students in relation to their attitudes, knowledge and expectations towards the oncology patient	Emphasizes the importance of social and communication skills in the interaction with the oncology patient and his/her family
Borneman et al. 2011 [19]	Quasi experimental	Evaluate the effectiveness of an educational intervention to reduce pain and fatigue in cancer patients	Reduced pain and fatigue barriers and improved cognition
Trevisani et al. 2015 [20]	Descriptive and qualitative	Identify whether the use of concept mapping strategy assists a student to extend and revise their expertise in oncology and analyze the abilities developed in a student in order to go through theoretical to practical knowledge	An increase in autonomy and clinical reasoning in nursing practice
Mitchell et al. 2019 [21]	Review	Evaluate a fourth-year undergraduate nursing oncology course at the University of Calgary	Need to provide more student-centered learning
Flocke et al. 2017 [22]	Descriptive	Describe the experiences of oncology nurses	Barriers in the treatment and counselling of patients (lack of knowledge)
Davis et al. 2017 [23]	Interpretive phenomenological approach	Know how oncology nurses invested in building relationships with patients and their family members	Were seen to make a difference in the lives of patients and their family members by supporting them through the cancer journey
Gustaffson et al. 2013 [24]	Quasi-experimental	Investigate without a theory-based educational intervention which can modify knowledge and attitudes toward pain management in the oncologic patient	The intervention program is effective in changing knowledge and perceptions of pain management
Santana-Padilla et al. 2019 [25]	Descriptive	Assess the training needs detected by ICU nurses	Inadequate prior training and need for continuous training and refresher training
Vila et al. 2016 [26]	Review	Define a new oncology nursing role to guide breast cancer patients	The new oncology nursing role seems useful in helping patients with problem solving, developing teamwork skills, and coping with burnout symptomatology
De Mata et al. 2017 [27]	Descriptive	Identify the feelings of nurses working with terminally ill cancer patients	One of the main difficulties is to accompany the patient in the dying process
Petersen et al. 2017 [28]	Descriptive	Describe and evaluate an online program for pediatric oncology nurses to help improve knowledge, attitudes and professional competence in spiritual care	The effectiveness of the program is confirmed
Poulsen et al. 2015 [29]	Randomized clinical trial	Evaluate the effectiveness of an educational intervention to reduce the effects of stress in oncology nurses	Improvement after program implementation
Jors et al. 2015 [30]	Descriptive	Explore through a program on oncology nurses’ perception of end-of-life care in terminally ill patients	The need for continuous training in this area throughout professional life is evident
Banerjee 2017 [31]	Quasi-experimental	Evaluate the efficacy of a communication skills-based oncology program for nurses	Improved self-efficacy and safety of program participants
Stanciu et al. 2015 [32]	Randomized clinical trial	Evaluate oncology nursing interventions using the needs assessment tool	Effectiveness of the program
James et al. 2016 [33]	Qualitative	Develop interprofessional team training opportunities using simulated cancer care scenarios to enhance collaborative practice skills within clinical oncology	These types of training programs have the potential to transform cancer care by creating high-performing teams resulting in improved patient outcomes
Korber et al. 2011 [34]	Descriptive	Identify barriers to nursing care through a nursing care program	The importance of education and specific training in oncology is recognized
Santamaría et al. 2015 [35]	Review	Describe the state of the art on the perception of nursing care to oncology patients	Strategies for measuring and qualifying nurse-patient interaction are needed
Dos Santos et al. 2018 [36]	Descriptive	Describe continuing education strategies in oncology nursing	The importance of continuing education in oncology is emphasized
Zaider et al. 2018 [37]	Quasi-experimental	Evaluate an oncology nursing intervention program on the management of complex situations with patients and their families	Effectiveness of the program to improve the management of complex situations
Watson et al. 2014 [38]	Randomized clinical trial	Implement a psychoeducational program by oncology nurses in prostate cancer to find out the existing deficits in nursing care	The importance of addressing the psychological and physical needs of patients and of continuing education is noted
Schwappach et al. 2014 [39]	Qualitative	Explore factors that affect oncology staff’s decision to voice safety concerns or to remain silent and to describe the trade-offs they make	There is in-depth insight into fears and conditions conducive of silence and voicing and can be used for educational interventions and leader reinforcement
Domínguez-Nogueira et al. 2007 [40]	Qualitative	Determine how difficulties in communicating with cancer patients and their relatives are perceived by staff in oncology departments	Training in emotional and communication skills should be provided
Medina 2012 [41]	Integrative review	Determine if internal communication can be considered a managerial tool with strategic value	Internal communication is a true managerial tool because it affects all hospital employees and influences the centers’ operations
Eid et al. 2009 [42]	Quasi-experimental	Evaluate the effectiveness of an intervention program aimed at improving communication skills in the hemato-oncology patient	Utility of intervention to improve bad news communication skills
Cara et al. 2018 [46]	Descriptive	Understand the active listening needs of oncology nurses and their patients	The difficulty of communicating with this type of patient is recognized
Rohani et al. 2020 [47]	Descriptive	Explore the importance of empathy in oncology patients from the nurses’ point of view	Clinical empathy should be considered as one of the essential competencies for nurses
Leal-Costa et al. 2020 [48]	Quasi-experimental	Provide new insights regarding the effects of nursing communication skills on perceived self-efficacy	Communication skills can positively impact self-efficacy
Mojrad et al. 2019 [49]	Descriptive	Determine facilitators and barriers to oncology patient management	Categories: spirituality, compassion and effective communication
Badger et al. 2013 [50]	Quasi-experimental	Analyze a sample of cancer survivors to determine which of two interventions was more effective in improving psychological quality of life	Low consistency due to small sample size
Wittenberg et al. 2020 [51]	Descriptive	Review nursing communication strategies with palliative oncology patients through the COMFORT program	Communication skills essential in oncology nursing
Pongthavornkamol et al. 2018 [52]	Descriptive	Explore a group of nurses’ perceptions of the oncology patient during hospitalization	Sometimes nurses does not involve the patient in the care process
Harden et al. 2014 [53]	Quasi-experimental	Improve palliative care nurses’ knowledge, attitudes and behaviors after a training program	Confirmation of improved knowledge, attitudes and behaviors of palliative care nurses after the training program
Park et al. 2018 [54]	Descriptive	Develop, implement and evaluate a training program for healthcare professionals to improve capacity in care delivery	Effectiveness of a psychosocial support program for cancer survivors
Thomas et al. 2012 [55]	Randomized clinical trial	Evaluate two interventions compared with traditional intervention to minimize attitudinal barriers to pain management in the oncology patient	Motivational approach to pain management in oncology patients emerges as a useful approach
Henoch et al. 2013 [56]	Randomized clinical trial	Determine the effects of a training intervention focused on the perception of nursing safety in communication with oncology patients	Improved assurance of nursing care in staff nurses in relation to communication
Hendricks-Jackson et al. 2017 [57]	Randomized clinical trial	Determine the effectiveness of a psychoeducation program in oncology nursing	Reduction in psychological distress in patients
Malone et al. 2007 [58]	Quasi experimental	Alleviate some of the anxiety associated with the first chemotherapy experience	Effectiveness of the program is demonstrated
Wright et al. 2012 [59]	Review	Stimulate reflection amongst oncology nurses and nursing leaders	The McGill Model of Nursing with reference to how its ideas can support nursing practice for patients with cancer during the end-of-life phase
De Leeuw et al. 2013 [60]	Review	Provide input and opinion for future research, clinical practice development and nursing leadership	There are services that provide high-quality care that is both safe and efficient
Schofield et al. 2016 [61]	Randomized clinical trial	Assess the benefits of a nursing intervention on undetected patient needs and quality of life in patients with prostate cancer	A slight decrease in the depression of patients is noted
Tuominen et al. 2019 [62]	Overview of systematic reviews	Explore nursing interventions used among patients with cancer and summarize the results of their effectiveness	Nursing interventions achieved significant physical and psychological effects on the lives of patients with cancer
Martínez et al. 2015 [63]	Quasi-experimental	Examine differences in utilization of health services in oncology patients following nursing intervention	Psychoeducational intervention does not influence the utilization of health services
Mahendran et al. 2015 [64]	Quasi experimental	Evaluate the efficacy of a brief psychosocial intervention program in cancer patients	Patients improve their levels of depression, anxiety and increase their quality of life after the intervention
Prieto-Agüero et al. 2016 [65]	Review	Understand the communication training needs of oncology nurses	Adequate training for nursing professionals improves their safety in the treatment of patients

## Data Availability

Not applicable.

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
