# Peer review of "Educational Needs in Oncology Nursing: A Scoping Review"

_healthcare, 2022, doi:10.3390/healthcare10122494_

Round 1

Reviewer 1 Report

The aim of this study is to design a training program to improve the communication skills of oncology nurses.

The study is quite comprehensive, combining a systematic review methodology with a SWOT analysis to detect the weaknesses and strengths of the literature reviewed to complete the proposal.

The results and discussion are comprehensive.

The main weakness is in the conclusions, which are rather "implications for practice" or "some final considerations".

I REcommend writing up the conclusions of the study, some of them are in the discussion.

Author Response

Reviewer 1

The aim of this study is to design a training program to improve the communication skills of oncology nurses.

The study is quite comprehensive, combining a systematic review methodology with a SWOT analysis to detect the weaknesses and strengths of the literature reviewed to complete the proposal.

The results and discussion are comprehensive.

The main weakness is in the conclusions, which are rather "implications for practice" or "some final considerations".

I recommend writing up the conclusions of the study, some of them are in the discussion.

We would like to thank Reviewer 1 for the comment, which we appreciate and proceed to modify the conclusions, since what it is suggested is more adequate.

Reviewer 2 Report

This manuscript is an interesting contribution to the topic of advanced nursing education. However, there are some inaccuracies listed below:

- The methodology used is not that of the systematic review, but that of the scoping review, therefore it is necessary to replace this sentence throughout the text (including title).

- Replace the word "training" with "education" throughout the text, if and where appropriate (including title); Education is defined as the process of gaining knowledge, skill, and development from study or training. Training is defined as the process of learning the skills one needs to do a particular job or activity.

- Table 1: in the search strategies, the terms "breast feeding" and "feeding" are reported, which are not pertinent to the topic under study. A typo? Please verify.

- Figure 1: check that the numbers shown in the flow diagram are correct. In addition, included study designs were erroneously reported as if they were inclusion criteria; please verify.

I remain at your disposal for further clarifications.

Best regards

Author Response

This manuscript is an interesting contribution to the topic of advanced nursing education. However, there are some inaccuracies listed below:

- The methodology used is not that of the systematic review, but that of the scoping review, therefore it is necessary to replace this sentence throughout the text (including title).

We thank Reviewer 2 for this valuable observation, we have replaced systematic review for scoping review as suggested.

- Replace the word "training" with "education" throughout the text, if and where appropriate (including title); Education is defined as the process of gaining knowledge, skill, and development from study or training. Training is defined as the process of learning the skills one needs to do a particular job or activity.

As recommended, the adequate changes have been made to correct “Training” for “Education” where appropriate.

- Table 1: in the search strategies, the terms "breast feeding" and "feeding" are reported, which are not pertinent to the topic under study. A typo? Please verify.

Regarding Table 1, thanks for this important observation. It was mistakenly copy-pasted. Apologies

- Figure 1: check that the numbers shown in the flow diagram are correct. In addition, included study designs were erroneously reported as if they were inclusion criteria; please verify.

As per Figure 1, the numbers are correct. However, we have erased a sentence in the text that leads to confusion. Moreover, we have swapped Inclusion Criteria for Types of Studies, as it was certainly incorrect. Thank you very much once again